# FAST READING COMPREHENSION WITH CONVNETS

## ABSTRACT

State-of-the-art deep reading comprehension models are dominated by recurrent neural nets. Their sequential nature is a natural fit for language, but it also precludes parallelization within an instances and often becomes the bottleneck for deploying such models to latency critical scenarios. This is particularly problematic for longer texts. Here we present a convolutional architecture as an alternative to these recurrent architectures. Using simple dilated convolutional units in place of recurrent ones, we achieve results comparable to the state of the art on two question answering tasks, while at the same time achieving up to two orders of magnitude speedups for question answering.

## 1 INTRODUCTION

Recurrent neural networks (e.g., LSTMs (Hochreiter & Schmidhuber, 1997) and GRUs (Cho et al., 2014)) are been very successful at simplifying certain natural language processing (NLP) systems, such as language modeling and machine translation. The dominant of *text understanding* (Rajpurkar et al., 2016; Joshi et al., 2017; Seo et al., 2017) typically relies on recurrent networks to produce initial representations for the question and the document, and then apply attention mechanisms (Bahdanau et al., 2014) to allow information passes between the two representations. The recurrent units are powerful structures capable of modeling complex long range interactions. However, their sequential nature precludes parallelization within training examples, and often become the bottleneck for deploying models to latency critical NLP applications. High latency is especially critical for interactive question answering (for example as part of search engines or mobile assistants), as it requires the user to wait patiently for the answer.

Recent development of "attention only" models Parikh et al. (2016); Vaswani et al. (2017) in various tasks, allows modeling of long range dependencies without regard to their distance. By parallelization within one instance, these models can have much better inference time than those which depend on recurrent units. However, their token-pair based attention requires $O(n^2)$ memory consumption within the GPUs, where $n$ denotes the length of the document. This quadratic growth prevents their use with most real-world documents, such as e.g. Wikipedia pages.

In this work we propose, Gated Linear Dilated Residual Network(GLDR), a different architecture to avoid recurrent units in text precessing. More specifically, we use a combination of residual networks (He et al., 2016), dilated convolutions (Yu & Koltun, 2016) and gated linear units (Dauphin et al., 2017).

### 1.1 READING COMPREHENSION TASKS

Reading comprehension tasks focus on one's ability to read a piece of text and subsequently answer questions about it (see Trival QA examples in Figure 1). We follow the typical reading compression setting and assume that the correct answer can be given as a snippet of the original text. This reduces the problem to a search problem, where the question functions as a query. Sometimes these tasks involve easy type and query term matching (e.g., answering *what philosopher taught Plato and Aristophanes?* mainly involves matching the entity names and a keyword); but they can also contain difficult language understanding (e.g., answering *who was the choreographer of the dance troupe Hot Gossip?* may involve the understanding that *dance troupe* is less informative than *Hot Gossip* for finding the correct answer.) and even real world knowledge (e.g., answer *What is the next in the series: Carboniferous, Permian, Triassic, Jurassic?* potentially involves understanding the semantics of *next*, and applying it on a structured data such as a knowledge base). Figure 2 shows an

> *Q: What is the next in the series: Carboniferous, Permian, Triassic, Jurassic?*
> ... The Jurassic North Atlantic Ocean was relatively narrow , while the South Atlantic did not open until the following **Cretaceous** period , when ...
>
> *Q: Plato and Xenophon were both pupils of which Greek philosopher?*
> **Socrates** ... philosophy. He is an enigmatic figure known chiefly through the accounts of classical writers, especially the writings of his students Plato and Xenophon and the plays of his contemporary Aristophanes. ...
>
> *Q: Who was the choreographer of the dance troupe Hot Gossip?*
> **Arlene Phillips**, ... Lee Mack. Hot Gossip In Britain, Phillips first became a household name as the director and choreographer of Hot Gossip, a British dance troupe which she formed in 1974 ...

Figure 1: Samples from Trivia QA (Joshi et al., 2017)

adversary example of a question that is answered incorrectly by matching the first occurrence of the query word "composed" in the answer text. This study will use two popular reading comprehension tasks – Trivia QA (Joshi et al., 2017), and SQUAD (Rajpurkar et al., 2016) – as its test bed. Both tasks have openly available training and validation data sets and are associated with competitions over a hidden test set on a public leaderboard.

Because of the sequential nature of documents and text, and complex long-distance relationships between words, recurrent neural networks (especially LSTMs Hochreiter & Schmidhuber (1997) and GRUs Cho et al. (2014)) are a natural class for modeling reading comprehension. Indeed, on both reading comprehension tasks we study here, every published result on the leader-board [1][2] uses some kind of recurrent mechanism. Below, we will discuss two specific models in detail, but we begin by motivating a one-dimensional convolution architecture for the reading comprehension task.

## 1.2 TEXT UNDERSTANDING WITH DILATED CONVOLUTIONS

Our goal is to substitute complicated and costly sequential models through simple feed-forward network architectures. Bidirectional recurrent units can in theory model arbitrarily long dependencies in text, but in practice we may be able to capture these dependencies through other mechanisms. There are two important criteria of language that LSTMs model, that we also want to capture. First, we may need to model relationships between individual words, even when they are separated from each through many words (e.g. Figure 1). Second, we want to model the compositional nature of natural language semantics, where the meaning of large phrases are composed of the meaning of their sub-phrases.

These constraints lead us to choose dilated convolutional networks (Yu & Koltun, 2016) with gated linear units (Dauphin et al., 2017). By increasing the receptive field in our convolutional units, dilation can help to model arbitrarily long-distance dependencies. Unfortunately, the receptive region is pre-determined, which prevents us from examining long range dependencies in detail. For instance,

---

[1] https://competitions.codalab.org/competitions/17208
[2] https://rajpurkar.github.io/SQuAD-explorer

> *Q: Who composed the works The Fountains of Rome and The Pines of Rome in 1916 and 1924 respectively?*
> Adversary example . Wolfgang Amadeus Mozart composed The Magic Flute , and Requiem . **Ottorino Respighi** ( ; 9 July 1879 - 18 April 1936 ) was an Italian violinist , composer and musicologist , best known for his three orchestral tone poems Fountains of Rome ( 1916 ) , Pines of Rome ( 1924 ) , and Roman Festivals ( 1928 ).

Figure 2: DrQA gives a wrong answer "Wolfgang Amadeus Mozart" rather than "Ottorino Respighi" by simply matching the verb "composed".

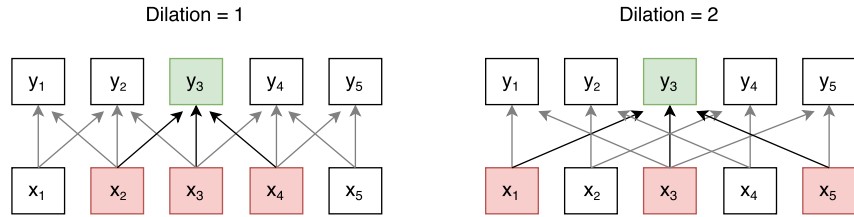

Figure 3: An illustration of dilated convolution. With a dilation of 1 (*left*), dilated convolution reverts to standard convolution. With a dilation of 2 (*right*) every other word is skipped, allowing the output $y_3$ to relate words $x_1$ and $x_5$ despite their large distance and a relatively small convolutional kernel.

| Layer type | Computations per layer | Minimum depth $D$ to cover length $n$ | Longest computation path | Overall computations |
|---|---|---|---|---|
| Recurrent Units | $O(d^2n)$ | $O(1)$ | $O(nD)$ | $O(d^2n)$ |
| Self-Attention | $O(dn^2)$ | $O(1)$ | $O(D)$ | $O(dn^2)$ |
| Dilated Convolution | $O(kd^2n)$ | $O(\log(n))$ | $O(D)$ | $O(kd^2nD)$ or $O(kd^2n\log(n))$ |

Table 1: Comparison among three sequence encoding layers with input sequence length $n$, network width $d$, kernel size $k$, and network depth $D$. Recurrent units and self-attention become slow as $n$ grows. When the receptive field of a dilated convolution covers the longest possible sequence $n$, its overall computation is proportional to $O(\log n)$.

in the co-reference example from Figure 1, we would need to directly convolve representations for "his" and "Socrates", but an increasing dilation will miss this. In practice, "Socrates" is combined with its context to give a fixed size representation for a long context. We alleviate some of this effect by using Gated Linear Units (Dauphin et al., 2017) in our convolutions. These units allow us to selectively retain (and compute gradients for) important features of low-level words and phrases, even at convolutions with larger dilations.

**Dilated Convolution.** Given a 1-D convolutional kernel $\mathbf{k} = [k_{-l}, k_{-l+1}, ..., k_l]$ of size $2l - 1$ and the input sequence $\mathbf{x} = [x_1, x_2, ..., x_n]$ of length $n$, a $d$ dilated convolution of $\mathbf{x}$ with respect the kernel $\mathbf{k}$ can be described as

$$(\mathbf{k} * \mathbf{x})_t = \sum_{i=-l}^{l} k_i \cdot x_{t-d \cdot i}$$

where $t \in \{1, 2, \cdots, n\}$. Here we assume zero-padding, so tokens outside the sequence will be treated as zeros. Unlike normal convolutions (i.e. $d = 1$) that convolve each contiguous subsequence of the input sequence with the kernel, dilated convolution uses every $d^{th}$ element in the sequence, but shifting the input by one at a time. Figure 3 shows an example of dilated convolution. Here, the green output is a weighted combination of the red input words.

**Why Dilated convolution?** Repeated dilated convolution (Yu & Koltun, 2016) increases the receptive region of ConvNet outputs exponentially with respect to the network depth, which results in drastically shortened computation paths. See Figure 4 for an illustration of an architecture with four dilated convolutional layers with exponentially increasing dilations. Table 1 shows a brief comparison between bidirectional recurrent units, self-attention, and dilated convolution. Self-attention suffers from the fact that the overall computation is quadratic with respect to the sequence length $n$. This may be tolerable in settings like machine translation, where a typical document consists of less than $n < 100$ words and a wide network is often used (i.e. $d$ is large); however, for reading comprehension tasks, where long documents and narrow networks are typical (i.e. $n \gg d$), self-attention becomes expen-

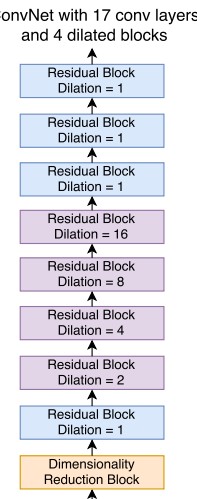

Figure 4: The receptive field of repeated dilated convolution grows exponentially with network depth.

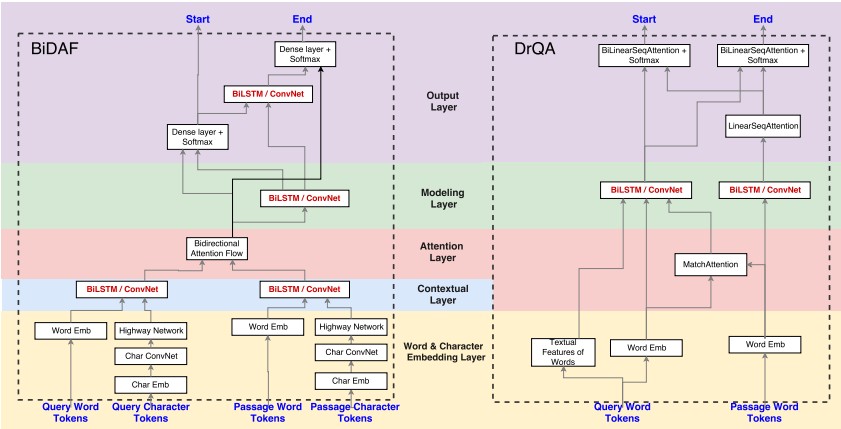

Figure 5: Schematic layouts of the BiDAF (*left*) and DrQA (*right*) architectures.

sive. In addition, bidirectional recurrent units have the intrinsic problem that their sequential nature precludes parallel processing.

Admittedly, dilation has its limitations. It requires more overall computations than recurrent nets and the reception region is predetermined. We argue that to provide answers as a web service, one cares more about the response latency for a single question. Therefore, a short compute of the longest computation is favored.

## 1.3 BASELINE MODELS: BIDAF AND DRQA

In the following we briefly describe two popular open-sourced question answering systems: Bidirectional Attention Flow (BiDAF) (Seo et al., 2017) and DrQA (Chen et al., 2017a), which are relevant to our study. Figure 5 shows schematic layouts of their respective model structures and highlight the BiLSTM layers (in red) which are the bottleneck for inference speed. Both models require LSTMs to encode the query and passage. BiDAF is more complex than DrQA, with two more LSTMs to make the classification decision.

## 1.4 RELATED WORK

**The BiDAF** model (Seo et al., 2017) has six components: 1) character embedding layer, 2) word embedding layer, 3) contextual layer, 4) attention flow layer, 5) modeling layer, and 6) output layer, but only three of them contains LSTMs. The contextual layer encodes the passage and the query with two bidirectional LSTMs with shared weights. The modeling layer further employs a two-layer stacked bidirectional LSTM to extract the higher order features of the words in the passage. The output layer uses yet another bidirectional layers to produces features for predicting the end of the answer span.

**The DrQA** system (Chen et al., 2017a) has a document retriever and a document reader. The document retriever simply uses pre-defined features to retrieve documents when the corresponding passage is not given in the question. The document reader uses two 3-layer stacked bidirectional LSTMs to encode the query and the passage/document respectively.

**ConvNets for Text** There has been a lot of effort in applying ConvNet architecture to reduce the sequential computation in sequence to sequence models such as Extended Neural GPU(Kaiser & Bengio, 2016), ByteNet (Kalchbrenner et al., 2016) and ConvS2S Gehring et al. (2017) In these models, the number of operations required to relate signals from two arbitrary input or output positions grows in the distance between positions, linearly for ConvS2S and logarithmically for ByteNet.

The improvements in speed are limited in these generative models, because the decoding procedure still needs to be done token by token. In comparison, there is no generation in reading comprehension models (Seo et al., 2017; Chen et al., 2017a), therefore much more impressive speedups are possible.

**Reading Comprehension Models** After the release of the Stanford Question Answering Dataset (Rajpurkar et al., 2016), reading comprehension models kept springing up in the past year. All of them use recurrent neural networks (Hochreiter & Schmidhuber, 1997; Cho et al., 2014) as a common component, and most of the top performed models uses attention (Bahdanau et al., 2014) in addition. DCN proposed to have interactions between question and answer multiple times. BiDAF (Seo et al., 2017)introduced a bidirectional attention flow to incorporate the passage with query and was the first open sourced state-of-the-art model at the time, RasorNet proposed to estimate the joint distribution of the answer span directly rather than modeling the start and end points independently. RNet, ..., demonstrates the effectiveness of the self-attention modules. Document Reader (DrQA) provides an question answering system using a document database. Hu et al. (2017) demonstrated how reinforcement learning can benefits the training procedure. Smarnet proposed to mechanism to keep refining the prediction.

## 2 QUESTION ANSWERING WITH STRIDED CONVOLUTIONAL

### 2.1 GATED LINEAR DILATED RESIDUAL NETWORK (GLDR)

Our proposed GLDR structure contains a dimensionality reduction block followed by a few residual blocks (shown in Figures 6 and 7 respectively). The dimensionality reduction block comprises a dropout layer (Hinton et al., 2012), a normal convolution (kernel size 3) and a gated linear unit (GLU) as activation. A residual block has a two-layer ConvNet with input dropout before the convolutions and GLU activations. The output of this small two-layer ConvNet is later summed up with the input allowing the convnet to learning only the residual of the transformation. For simplicity all the convolutions are of kernel size 3 while the dilations vary across layers. To be more specific, the dilations of the convolutions in the first few residual blocks are increased exponential $(1, 2, 4, 8, \cdots)$, see Figure 4 for an illustration. The purpose of the dilated convolution is to increase the receptive field. After a small number of layers $O(\log(n))$, the receptive field is wide enough and we switch back to the use of normal convolutions for the rest of the blocks.

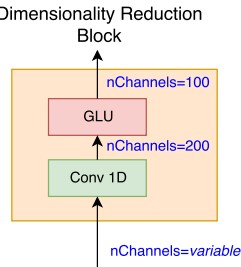

Figure 6: Dimensionality reduction through convolution.

To show ConvNets are not limited to a particular architecture of question answers models, we apply our method to two popular open-sourced question answering systems: Bi-directional Attention Flow (BiDAF) (Seo et al., 2017) and DrQA (Chen et al., 2017a).

### 2.2 CONVOLUTIONAL BIDAF

In our convolutional version of BiDAF, we replaced all bidirectional LSTMs with ConvNets. We have two 5-layer ConvNets in the contextual layer whose weights are un-tied because we saw a performance gain. In the modeling layer, a 17-layer ConvNets with dilation 1, 2, 4, 8, 16 in the first 5 residual blocks is used, which results in a reception region of 65 words. A 3-layer ConvNet replaces the bidirectional LSTM in the output layer. For simplicity, we use same-padding and kernel size 3 for all convolutions unless specified. The hidden size of the ConvNet is 100 which is the same as the LSTM in BiDAF.

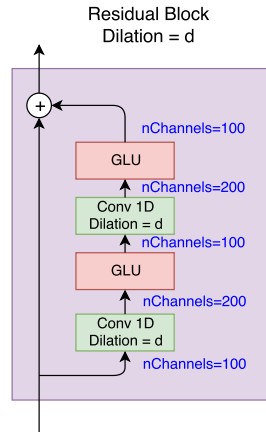

Figure 7: Dimensionality preserving residual block.

### 2.3 CONVOLUTIONAL DRQA

Since the query is much shorter than the document, we can afford using a 17-layer ConvNet without dilation for encoding the query. On the other hand, a 9-layer ConvNet whose 4 residual blocks have dialtion 1, 2, 4, and 8, respectively is used to capture the context information in the passage, which results a reception region of 33 words. The hidden size of the ConvNet is 128 which is the same as the LSTM in DrQA.

## 3 EXPERIMENTS

### 3.1 STANFORD QUESTION ANSWERING DATASET (SQUAD)

The Stanford Question Answering Dataset or SQuAD (Rajpurkar et al., 2016) is one of the most popular reading comprehension datasets, which contains more than 100K questions-answer-passage tuples labeled by crowdsource workers. Given a passage a group of workers were asked to generate questions based on it, and another group of workers attempted to highlight a span in the passage as the answer. This ensures that the passage also contains sufficient information and the answer is always there.

We show that Conv BiDAF achieves comparable results to BiDAF on SQuAD development set while being much efficient during both training and inference. On the other hand, the Conv DrQA inferences even faster than Conv BiDAF while sacrificing the performance.

**Experiment Setup** We adopted the open-sourced BiDAF implementation[3] which is written in TensorFlow (Abadi et al., 2016) and DrQA implementation[4] in PyTorch [5], and followed their preprocessing and experiment setup. We conducted experiments on the SQuAD validation set since we were not aiming at the state-of-the-art performance. The models were trained with either a NVIDIA Tesla P100 GPU or a NVIDIA Titan X (Pascal) GPU, but the timing experiments were only conducted on a single Titan X (Pascal) GPU. We only timed GPU eclipsed time (i.e. forwarding and backwarding through the networks) in all experiments since CPU bounded operations are not our focuses.

For BiDAF, we used batch size 60, dropout rate 0.2 (Srivastava et al., 2014), and trained the model for 20000 as the default setting in BiDAF. In addition, we used stochastic gradient descend with momentum 0.1 and weight decay $10^{-4}$, which we found improved the performance of BiDAF slightly. For our Conv BiDAF, we trained the model for 60000 with Adam optimizer (Kingma & Ba, 2014) using the default settings in TensorFlow ($\alpha = 0.0001, \beta_1 = 0.9, \beta_2 = 0.999$), dropped $\alpha$ by a factor of 10 every 20000 iterations, and used an additional word dropout rate 0.1 (Dai & Le, 2015). Word dropout wasn't found helpful for BiDAF in our experiment. Because of the GPU memory constraint, the model was trained with the documents shorter than or equal to 400 word tokens as what the authors did in the paper.

For all DrQA variants, we adopted batch size 32, dropout rate 0.3, and trained both models for 60 epochs with Adamax (Kingma & Ba, 2014) optimizer using the default setting in PyTorch ($\alpha = 0.002, \beta_1 = 0.9, \beta_2 = 0.999$). Weight decay and word dropout didn't result in a fair amount of improvement on either of the models, so they were abandoned in the reported models. The models were trained on the SQuAD training set without removing long documents.

**Results** As we can see in Figure 8, Conv BiDAF achieved one to two order of magnitude speed-up during training and inference and performed as well as BiDAF. More detailed comparison between BiDAF and Conv BiDAF was shown in Table 2.

On Table 3, we show different variants of BiDAF models and their performance.

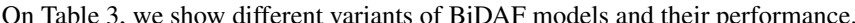
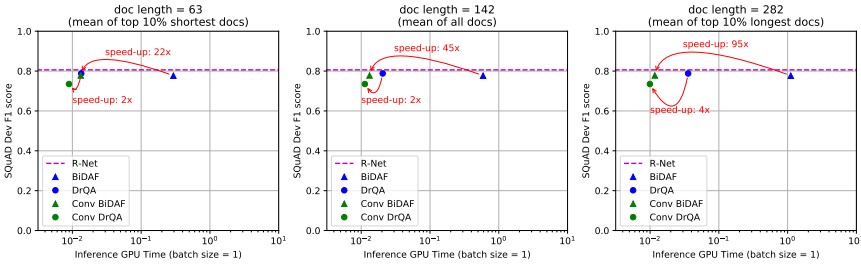

Figure 8: Dev F1 Score on SQuAD vs Inference GPU Time

---

[3]https://github.com/allenai/bi-att-flow
[4]https://github.com/facebookresearch/DrQA
[5]http://pytorch.org/

| Model | BiDAF | Conv BiDAF (5-17-3 conv layers) |
|---|---|---|
| # of params | 2.70M | 2.76M |
| Dev EM (Multiplied by 100) | $67.66 \pm 0.41$ | $68.28 \pm 0.56$ |
| Dev F1 (Multiplied by 100) | $77.29 \pm 0.38$ | $77.27 \pm 0.41$ |
| Training time (h) (1 GPU, until convergence [7]) | 21.5 | 3.4 (6.3x) |
| Training time per iteration (sec), batch size = 60, 1 GPU | $3.88 \pm 2.14$ | $0.204 \pm 0.135$ (19x) |
| Inference time per iteration (sec), batch size = 60, 1 GPU | $1.74 \pm 0.75$ | $0.0808 \pm 0.0021$ (21x) |
| Inference time per iteration (sec), batch size = 1, 1 GPU | $1.58 \pm 0.058$ | $0.0161 \pm 0.0026$ (98x) |

Table 2: BiDAF v.s. Conv BiDAF. For timing, we only reported the GPU time. EM stands for exact match score.

| Model | # of params | Dev EM | Dev F1 |
|---|---|---|---|
| BiDAF (trained by us) | 2.70M | 67.94 | 77.65 |
| Conv BiDAF (5-17-3 conv layers) | 2.76M | 68.87 | 77.76 |
| Conv BiDAF (9-9-3 conv layers) | 2.76M | 67.79 | 77.11 |
| Conv BiDAF (0-31-3 conv layers) | 3.36M | 63.52 | 72.39 |
| Conv BiDAF (11-51-3 conv layers) | 5.53M | 69.49 | 78.15 |
| Conv BiDAF (31-31-3 conv layers) | 6.73M | 68.69 | 77.61 |
| DrQA (trained by us) | 33.82M | 69.85 | 78.96 |
| Conv DrQA (9 conv layers) | 32.95M | 62.65 | 73.35 |

Table 3: Comparing variants with different number of layers. EM stands for exact match score. The scores are multiplied by 100. DrQA uses a much larger pre-trained word embedding resulting in more parameters.

## 3.2 TRIVIAQA

TriviaQA is new large-scale reading comprehension dataset with 95K question-answer pairs and 650K question-answer-evidence tuples which is more challenging than SQuAD because it 1) contains more complex questions, 2) has substantial syntactic and lexical variability in the text, 3) requires a significant amount of cross-sentence reasoning, and 4) the answer and the sufficient information are guaranteed in the evidence. For each question-answer pair, it used distant supervision to provide relevant evidence from wikipedia or web search. Besides the full development and test set. A verified subset for each is also provided.

With the same hyperparamters used for SQuAD, our Conv DrQA outperformed all models reported in the published literatures while being on par with unpublished ones on the wiki split leader-board. Again, we can see a trade-off between performance and speed.

**Experiment Setup** We processed the data into SQuAD format with the script provided by Trivia QA[8]. Precisely, for each document in the candidate set of a question-answer pair, it produces a question-answer-passage pair for training as long as any of the answers appear in the first 800 tokens in the document. For evaluation, we truncated each document down to 1600 tokens and predict a span among them.

**Results** On Wikipedia split of TriviaQA, our proposed Conv DrQA is slight worse than our DrQA baseline which beats all previous models, it can still be on a par with the previous state-of-the-art performance of recurrent networks. The numbers are shown in Table 5 The Conv DrQA model only encode every 33 tokens in the passage, which shows that such a small context is enough most of the question.

---

[8]https://github.com/mandarjoshi90/triviaqa

| Model | Dev EM | Dev F1 |
|---|---|---|
| Conv BiDAF (5-17-3 conv layers) | 68.87 | 77.76 |
| without dilation | 68.15 | 76.99 |
| Using ReLU instead of GLU | 63.91 | 73.39 |

Table 4: Ablation Test. EM stands for exact match score. The scores are multiplied by 100.

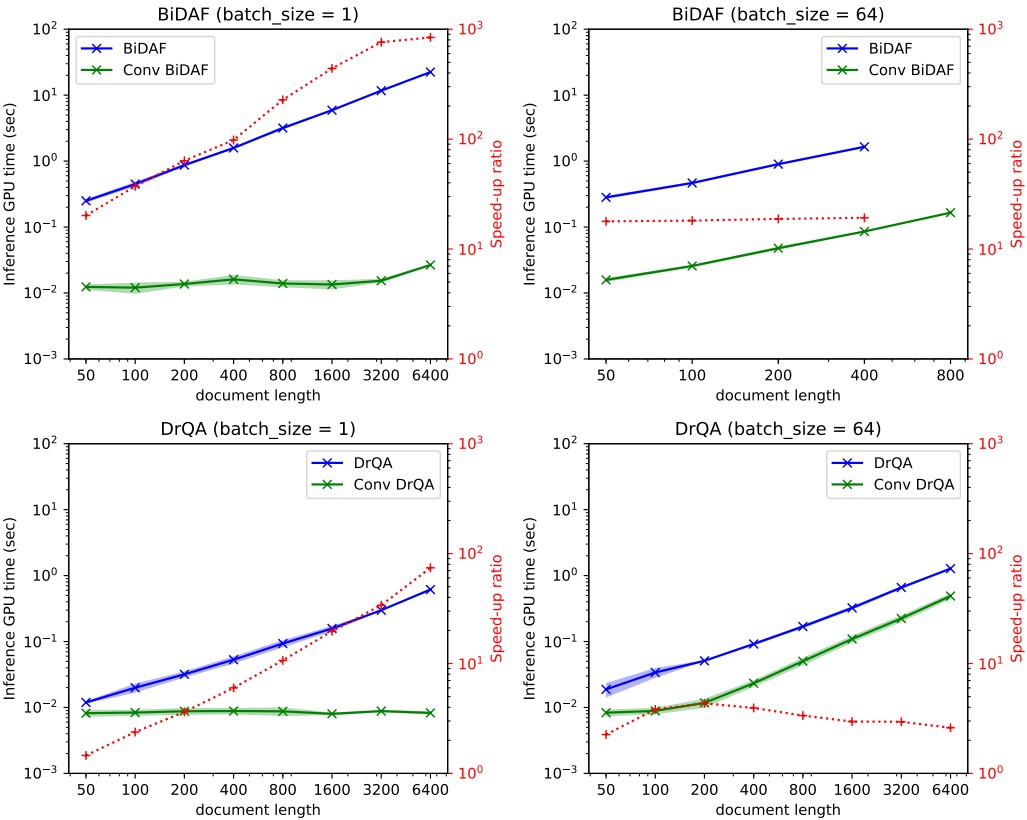

Figure 9: Inference GPU time of four models with batch size 1 or 64. The time spent on data pre-processing and decoding on CPUs are not included. We suspect the difference in speed-up is caused by implementation difference between TensorFlow and PyTorch. The missing points for BiDAF and Conv BiDAF were caused by running of out GPU memories.

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

## A   DETAILED PERFORMANCE COMPARISONS

| Dataset | Model | Full | | Verified | | Inference |
|---|---|---|---|---|---|---|
| | | EM | F1 | EM | F1 | GPU Time per instance |
| Wikipedia | BiDAF (Joshi et al., 2017) | 40.32 | 45.91 | 44.86 | 50.71 | |
| | RMR (Hu et al., 2017) | 46.94 | 52.85 | 54.45 | 59.46 | |
| | Smarnet (Chen et al., 2017b) | 42.41 | 48.84 | 50.51 | 55.90 | |
| | Leader-board (unpublished) [9] | 48.64 | 55.13 | 53.42 | 59.92 | |
| | DrQA | 52.58 | 58.17 | 57.36 | 62.55 | 93.2 ms |
| | Conv DrQA | 49.01 | 54.52 | 54.11 | 59.90 | 8.7 ms |
| Web | BiDAF (Joshi et al., 2017) | 40.74 | 47.06 | 49.54 | 55.80 | |
| | RMR (Hu et al., 2017) | 46.65 | 52.89 | 56.96 | 61.48 | |
| | Smarnet (Chen et al., 2017b) | 40.87 | 47.09 | 51.11 | 55.98 | |
| | Leader-board (unpublished) | 50.56 | 56.73 | 63.20 | 67.97 | |
| | DrQA | 51.49 | 57.87 | 62.55 | 67.84 | 93.2 ms |
| | Conv DrQA | 47.77 | 54.33 | 57.35 | 62.23 | 8.7 ms |

Table 5: TriviaQA Performance. The scores are multiplied by 100.

