# OpenReview forum: "FAST READING COMPREHENSION WITH CONVNETS"
_ICLR.cc/2018/Conference — Reject_

### Official Review · AnonReviewer3 · 2017-11-23
**No comparison with the more straightforward method of sentence-based parallelization, resulting in weak motivation and contribution. The results are not very well generalized to other RC model like DrQA.**

**Rating:** 4
**Confidence:** 4

**Review:**

This paper borrows the idea from dilated CNN and proposes a dilated convolution based module for fast reading comprehension, in order to deal with the processing of very long documents in many reading comprehension tasks. The method part is clear and well-written. The results are fine when the idea is applied to the BiDAF model, but are not very well on the DrQA model.

(1) My biggest concern is about the motivation of the paper:

Firstly, another popular approach to speed up reading comprehension models is hierarchical (coarse-to-fine) processing of passages, where the first step processes sentences independently (which could be parallelized), then the second step makes predictions over the whole passage by taking the sentence processing results. Examples include , "Attention-Based Convolutional Neural Network for Machine Comprehension", "A Parallel-Hierarchical Model for Machine Comprehension on Sparse Data", and "Coarse-to-fine question answering for long documents"

This paper does not compare to the above style of approach empirically, but the hierarchical approach seems to have more advantages and seems a more straightforward solution.

Secondly, many existing works on multiple passage reading comprehension (or open-domain QA as often named in the papers) found that dealing with sentence-level passages could result in better (or on par) results compared with working on the whole documents. Examples include "QUASAR: Datasets for question answering by search and reading", "SearchQA: A new q&a dataset augmented with context from a search engine", and "Reinforced Ranker-Reader for Open-Domain Question Answering". If in many applications the sentence-level processing is already good enough, the motivation of doing speedup over LSTMs seems even waker.

Even on the SQuAD data, the sentence-level processing seems sufficient: as discussed in this paper about Table 5, the author mentioned (at the end of Page 7) that "the Conv DrQA model only encode every 33 tokens in the passage, which shows that such a small context is ENOUGH for most of the questions".

Moreover, the proposed method failed to give any performance boost, but resulted in a big performance drop on the better-performed DrQA system. Together with the above concerns, it makes me doubt the motivation of this work on reading comprehension.

I would agree that the idea of using dilated CNN (w/ residual connections) instead of BiLSTM could be a good solution to many online NLP services like document-level classification tasks. Therefore, the motivation of the paper may make more sense if the proposed method is applied to a different NLP task.

(2) A similar concern about the baselines: the paper did not compare with ANY previous work on speeding up RNNs, e.g. "Training RNNs as Fast as CNNs". The example work and its previous work also accelerated LSTM by several times without significant performance drop on some RC models (including DrQA).

(3) About the speedup: it could be imaged that the speedup from the usage of dilated CNN largely depends on the model architecture. Considering that the DrQA is a better system on both SQuAD and TriviaQA, the speedup on DrQA is thus more important. However, the DrQA has less usage of LSTMs, and in order to cover a large reception field, the dilated CNN version of DrQA has a 2-4 times speedup, but still works much worse. This makes the speedup less impressive.

(4) It seems that this paper was finished in a rush. The experimental results are not well explained and there is not enough analysis of the results.

(5) I do not quite understand the reason for the big performance drop on DrQA. Could you please provide more explanations and intuitions?

---

### Official Review · AnonReviewer2 · 2017-11-27
**interesting application of dilated convolution to replace recurrent networks**

**Rating:** 7
**Confidence:** 3

**Review:**

The paper proposes a simple dilated convolutional network as drop-in replacements for recurrent networks in reading comprehension tasks. The first advantage of the proposed model is short response time due to parallelism of non-sequential output generation, proved by experiments on the SQuAD dataset. The second advantage is its potentially better representation, proved by better results compared to models using recurrent networks on the TriviaQA dataset.

The idea of using dilated convolutional networks as drop-in replacements for recurrent networks should have more value than just reading comprehension tasks. The paper should stress on this a bit more. The paper also lacks discussion with other models that use dilated convolution in different ways, such as WaveNet[1].

In general, the proposed model has novelty. The experimental results also sufficiently demonstrate the proposed advantages of the model. Therefore I recommend acceptance for it.

[1] Oord, Aaron van den, Sander Dieleman, Heiga Zen, Karen Simonyan, Oriol Vinyals, Alex Graves, Nal Kalchbrenner, Andrew Senior, and Koray Kavukcuoglu. "Wavenet: A generative model for raw audio." arXiv preprint arXiv:1609.03499 (2016).

---

### Official Review · AnonReviewer1 · 2017-11-28

**Rating:** 5
**Confidence:** 4

**Review:**

This paper proposes a convnet-based neural network architecture for reading comprehension and demonstrates reasonably good performance on SQuAD and TriviaQA with a great speed-up.

The proposed architecture combines a few recent DL techniques: residual networks, dilated convolutions and gated linear units.

I understand the motivation that ConvNet has a great advantage of easing parallelization and thus is worth exploring. However, I think the proposed architecture in this paper is less motivated. Why is GLU chosen? Why is dilation used? According to Table 4, dilation is really not worth that much and GLU seems to be significantly better than ReLU, but why?

The architecture search (Table 3 and Figure 4) seems to quite arbitrary. I  would like to see more careful architecture search and ablation studies. Also, why is Conv DrQA significantly worse than DrQA while Conv BiDAF can be comparable to BiDAF?

I would like to see more explanations of Figure 4. How important is # of layers and residual connections?

Minor:
- It’d be helpful to add the formulation of gated linear units and residual layers.
- It is necessary to put Table 5 in the main paper instead of Appendix. These are still the main results of the paper.

---

### Decision · Program_Chairs · 2018-01-29
**ICLR 2018 Conference Acceptance Decision**

**Decision:**

Reject

**Comment:**

The key motivation for the work is producing both an efficient (parallelizable / fast) and accurate reading comprehension model. At least two reviewers are not convinced that this goal is really achieved (e.g., no comparison to hierarchical modeling, performance is not as strong).   I also share concerns of R1 that, without proper ablation search and more careful architecture choice, the modeling decisions seem somewhat arbitrary.

+ the goal (of achieving effective reading comprehesion models) is important
- alternative parallelization techniques (e.g., hierarchical modeling) are not considered
- ablation studies / more systematic architecture search are missing
- it is not clear that the drop in accuracy can be justified by the potential efficiency gains (also see details in R3 -> no author response to them)